



# Rapid warming and degradation of mountain permafrost in Norway and Iceland

Bernd Etzelmüller[1], Ketil Isaksen[1], Justyna Czekirda[2], Sebastian Westermann[2], Christin Hilbich[3], Christian Hauck[3]

[1] Department of Geosciences, University of Oslo, 1047 Blindern, Oslo, Norway
[2] Meteorological Institute of Norway, Oslo, Norway
[3] Department of Geosciences, University of Fribourg, Switzerland

*Correspondence to*: Bernd Etzelmüller (bernd.etzelmuller@geo.uio.no)

**Abstract.** With the EU-funded PACE project in the turn of this century, several deep boreholes (100 m +) were drilled in European mountain sites, including mainland Norway, Svalbard and Sweden. During other projects from c. 2004 and the International Polar Year (IPY) period in 2006/07, several additional boreholes were drilled in different sites in both Norway and Iceland, measuring temperatures along both altitudinal and latitudinal gradients. At most sites, multi-temporal geophysical soundings are available using seismic and electrical resistivity tomography (ERT). Here we study the development of permafrost and ground temperatures in mainland Norway and Iceland based on these data sets. We document that permafrost in is warming at an high rate, including the development of taliks in both Norway and Iceland in response to climate change during the last 20 years. At most sites ground surface temperature (GST) is apparently increasing stronger than surface air temperature (SAT). Changing snow conditions appear to be the most important factor for the higher GST rates. Modelling exercises also indicate that the talik development can by explained both by higher air temperatures and increasing snow cover.

## 1. Introduction

Permafrost is defined thermally as ground (i.e. lithosphere) at or below 0°C over at least two consecutive years (van Everdingen, 1998). Since the 18th century, permafrost has been known to be an important geomorphological factor governing certain landform development and producing geotechnical problems for construction (cf. French, 1996). Relatively recently, permafrost has been recognized as a major storage of carbon that can become mobilized and released as greenhouse gases upon thawing (Hugelius et al., 2014;Miner et al., 2022). Furthermore, permafrost is a major component for the stability of steep rock walls or debris





slopes in mountain environments (Gruber and Haeberli, 2007;Krautblatter et al., 2013;Penna et

al., 2023). Permafrost and the ground thermal regime also seem to be an important factor modulating geomorphological process rates (Berthling and Etzelmuller, 2011) and ultimately landscape development (Andersen et al., 2015;Egholm et al., 2015;Hales and Roering, 2007;Hales and Roering, 2009;Etzelmüller et al., 2020b).

Western Scandinavia and Iceland are situated at the transition zone between regions dominated

by mountain permafrost to Arctic conditions towards Svalbard and eastern Greenland. At present, Norway has an extensive network of boreholes where we measure subsurface temperatures along both altitudinal and latitudinal gradients (Etzelmüller et al., 2020a;Farbrot et al., 2011;Christiansen et al., 2010;Sollid et al., 2003). In addition, at most sites multi-temporal geophysical surveys are available using e.g. electrical resistivity tomography (ERT). In Iceland,

four boreholes exist since 2004, of which three were originally drilled in permafrost. Finally, daily gridded data sets of meteorological parameters such as air temperature and precipitation (Lussana et al., 2018a;Lussana et al., 2018b) and associated modelled snow cover (Saloranta, 2016;Czekirda et al., 2019) are available back to 1957 for Norway and 1959 for Iceland, allowing the evaluation of the relation between climate and ground thermal regime along

regional gradients.

This study outlines changes in the thermal state of permafrost in Norway and Iceland based on borehole monitoring between 2004 and 2022. The study demonstrates how the changing climate has rapidly warmed and degraded mountain permafrost and discusses the possible drivers for these changes.


## 2. Field sites and data

The field sites are located in five observatories in the mountain areas of southern and northern Norway, and around four boreholes in central and eastern Iceland (Fig. 1a, b). In Norway, all field sites are situated in typical mountain settings, with bedrock covered by relatively coarse-

grained regolith or glacial deposits. In Iceland, volcanic sand-rich deposits dominate the surface cover. All sites in Norway and Iceland are barren or only sparsely vegetated by lichen and mosses except the Iškoras site, which is covered by denser and higher vegetation. The geology varies between the sites, while the glaciation history is comparable. All sites were ice-covered during the last glaciations, however, most probably under cold basal ice conditions and thus

they experienced limited erosion at least during the last ice sheet period (e.g. Kleman and
    Hättestrand, 1999).

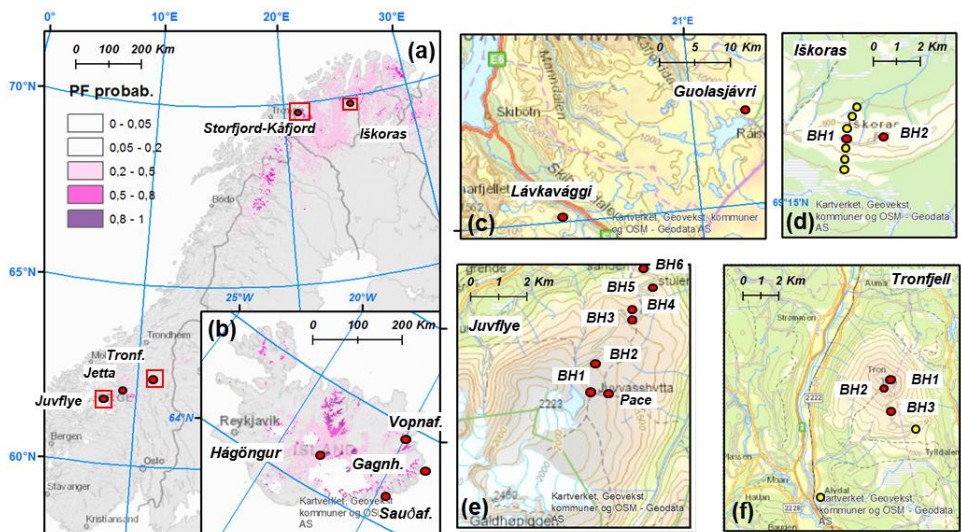

*Figure 1: Map of Norway (a) and Iceland (b), showing permafrost probability based on (Obu et al., 2019). The permafrost observatories are indicated with circles, and the close-up maps are indicated*
*with a rectangle. (c) The Storfjord-Kåfjord permafrost observatory (Troms county, northern Norway). (d) The Iškoras permafrost observatory (Finnmark county, northern Norway), (e) the Juvflye permafrost observatory (Innlandet county, southern Norway) and (f) the Tronfjell permafrost observatory (Innlandet county, southern Norway). Red dots denotes sites where we measured ground temperatures (GT), surface air temperatures (SAT) and ground surface temperatures (GST), while*
*yellow dots indicate only SAT and GST measurements at the site. All backround maps are © Norwegian Mapping Authority.*

## 2.1. The Juvflye permafrost observatory (Innlandet, southern Norway) (61.7°N, 8.4°E)

The Juvflye area is a high-mountain plateau at c. 1800 m a.s.l. which is surrounded by Norway's

highest peaks in Jotunheimen, with elevation close to 2500 m a.s.l. The bedrock is dominated

by metamorphosed gabbro, while the surface cover is dominated by blockfields and block-rich

ground moraines of some metres of thickness. In this area there are seven boreholes, of which

five are included in this study (Fig. 1e). They range from an elevation of 1500 m a.sl. to 1900

m a.s.l., of which the former is close to the lower altitudinal limit of permafrost in the area

(Hauck et al., 2004;Isaksen et al., 2002;Isaksen et al., 2011;Ødegård et al., 1996). The

uppermost boreholes are drilled in a blockfield-covered mountain plateau. The area has been

subject to long-term permafrost research (Farbrot et al., 2011;Hipp et al., 2012;Isaksen et al.,

2002;King, 1986;Ødegård et al., 1992) and has one of the deep (129 m) PACE boreholes



(Isaksen et al., 2001;Etzelmüller et al., 2020a) established in 1999. Long-term monitoring of air and ground surface temperatures takes place in addition to the borehole monitoring. The area also has intensive investigations on ice patches overlying permafrost (Ødegård et al., 2017).

## 2.2. The Tronfjell (62.2°N, 10.7°E) and Jetta (61.9°N, 9.3°E) permafrost observatory (southern Norway)

Tronfjell and Jetta are two mountain peaks, both at c. 1600 m a.s.l. and c. 50 km apart, which are easily accessible by a road. The Tronfjell mountain consists of a massif gabbro block, protruding the surrounding landscape. The mountain is surrounded by deep valleys at all sides

and therefore particularly prone to winter air temperature inversions. On Tronfjell three boreholes exists (Fig. 1f), of which we use the borehole at 1620 m a.s.l. located on the top plateau of the mountain massif (Farbrot et al., 2011) in this study. The Jetta mountain consists of metamorphosed schist, having two boreholes. Also here, we use the top borehole at 1580 m a.s.l.


## 2.3. The Storfjord-Kåfjord permafrost observatory (Troms, northern Norway)

The Storfjord-Kåfjord area in Troms comprises two different sites, Guolasjávri (69.4°N, 21.2°E) and Lávkavággi (69.3°N, 20.4°E), which are two neighbouring valleys, separated by a mountain range reaching up to c. 1600 m a.s.l. (Fig. 1c). The borehole at Guolasjávri is located

at c. 780 m a.s.l. on a mountain plateau close to the border to Finland, which is surrounded by peaks up to 1400 m a.s.l. The borehole at Lávkavággiis located at 770 m a.s.l. on a mountain pass between two valleys. At both sites the boreholes are located close to the lower limit of mountain permafrost, where snow thickness determines if a site develops permafrost or not (Christiansen et al., 2010;Farbrot et al., 2013).


## 2.4. The Iškoras permafrost observatory (Finnmark, northern Norway) (69.3°N, 25.3°E)

The Iškoras area consists of a quartzite massif protruding the peneplain of Finnmarksvidda, with a maximum elevation of 600 m a.s.l. There are two boreholes on the top of the Iškoras

Mountain, both at 600 m a.s.l. (Fig. 1 d). One borehole (I-BH1) is drilled directly into bedrock,




while borehole 2 (I-BH2) has a c. 3 m thick ground moraine cover over bedrock (Christiansen et al., 2010;Farbrot et al., 2013). In addition we measured air and ground surface temperatures along a transect in north-south direction over the ridge, between 600 m a.s.l. down to 200 m a.s.l. The plateau of the Finnmarksvidda undulates between 300 and 400 m a.s.l. The site is

frequently affected by winter air temperature inversions, especially below the tree line. Lakes and larger mire areas normally cover depressions on the Finnmarksvidda plateau. The area lies below the mountain permafrost belt, however, many of these mires contain palsas and large peat plateaus and were recently evaluated by Borge et al. (2017) and Martin et al. (2019).

| | Location | Elevation (in m) | BH depth (in m) | Drilled | Bedrock | Ground cover | Mean SAT (2007-2022) | Mean GST (2007-2022) | Mean GT_10 m (2007-2022) (trend, °C dec⁻¹) |
|---|---|---|---|---|---|---|---|---|---|
| Iskoras BH1 (Isk1) | 69.3°N 25.3°E | 585 | 10 | 2007 | Quartzite | Bedrock | same as BH2 | 0.5 °C | 0.5°C (+0.6) |
| Iskoras BH2 (Isk2) | 69.3°N 25.3°E | 591 | 58 | 2008 | Quartzite | Sandy/pebbly ground moraine | -1.2 °C | 0.7 °C | 0.2 °C (+0.6) |
| Lávkavággi (Lav1) | 69.15°N 20.3°E | 766 | 14 | 2007 | Schist | Bedrock | -2.0 °C | -0.5 °C | 0.0 °C |
| Guolasjavri BH1 GU1) | 69.4°N 21.2°E | 780 | 30 | 2007 | Schist | Bedrock | -1.8 °C | -0.6°C | 0.0 °C (+0.3) |
| Juvflye PACE (Juv-P) | 61.7°N 8.4°E | 1894 | 129 | 1999 | Gabbro | Regolith, Block field | -3.4°C | -2.8°C | -2.6°C (+0.2) |
| Juvflye BH1 (Juv1) | 61.7°N 8.4°E | 1851 | 10 | 2008 | Gabbro | Blocky ground moraine | -3.2°C | -2.8°C | -1.8°C (0.0) |
| Juvflye BH3 (Juv3) | 61.7°N 8.4°E | 1561 | 10 | 2008 | Gabbro | Ground moraine | same as BH4 | -0.4 °C | -0.6 °C (+0.5) |
| Juvflye BH4 (Juv4) | 61.7°N 8.4°E | 1547 | 15 | 2008 | Gabbro | Bedrock | -1.6 °C | -1.1 °C | -0.52 °C (+0.5) |
| Juvflye BH5 (Juv5) | 61.7°N 8.4°E | 1468 | 10 | 2008 | Gabbro | Ground moraine | -1.2 °C** | +0.1 °C*** | +1.1 °C (0.0) |
| Jetta BH1 (Jet1) | 61.9°N 9.3°E | 1560 | 12 | 2008 | Schists, sandstone (Precambrium) | Bedrock | -2.3 °C | 0.0 °C | -0.7 °C (+0.2) |
| Tronfjell BH1 (Tr1) | 62.2°N 10.7°E | 1640 | 30 | 2008 | Gabbro | Block field/ Blocky ground moraine | -2.7 °C | 0.7 °C | 0.1 °C (+0.4) |
| Hágöngur (Hag) | 64.6°N -18.3°E | 899 | 12 | 2004 | Basalt, Holcene | Sand, ash | -0.3 °C | 0.0 °C | 0.0 °C (+0.1) |
| Sauðafell (Sau) | 64.8°N, -15.6°E | 906 | 20 | 2004 | Basalt, Pleist. | Regolith, ash | -1.5 °C | -0.7°C | -0.4 °C (+0.2) |
| Vopnafjörður (*) (Vop) | 65.7°N -14.5°E | 892 | 22 | 2004 | Basalt, Upper Tert. | Regolith, morainic | -1.6 °C | 0.8 °C | 0.5°C (+0.3) |
| Gagnhaiði (Gag) | 65.2°N, -14.2°E | 931 | 14 | 2004 | Basalt, Uper Tert. | Regolith, morainic | -1.7 °C | -0.8 °C | -0.2 °C (+0.0) |

*Table 1: Borehole metadata and temperature trends during the measurement period. SAT = Surface air temperature, GST = ground surface temperature, GT = ground temperature, dec = decade, BH=borehole. *: GT from 20 m depth. **: the SAT station is located c. 100 m downslope of BH5, with an elevation of 1438 m asl. ***: The mean GST is calculated based on a nearby GST logger.*


### 2.5. The Iceland permafrost observatory (central and eastern Iceland)

The Iceland permafrost observatory (central and eastern Iceland) - Four boreholes were installed in 2004 in central (Hágöngur, 64.6°N, -18.3°E) and eastern Iceland (Sauðafell. 64.8°N, -15.6°E; Vopnafjörður, 65.7°N, -14.5°E; Gagnhaiði, 65.2°N, -14.2°E) (Fig. 1b). The boreholes

(8 - 20 m depth) are drilled in bedrock overlain by a sediment cover of c. 1 m. The surface cover consists of morainic deposits (Gagnhaiði) or vitrisols. This soil cover is poorly vegetated, where dry conditions prevail owing to: (1) sand-dominated sediments with low water holding capacity



and high hydraulic conductivity, (2) rapid evaporation during sunny spells in summer, when the dark surfaces heat up, and (3) limited infiltration during winter because of impermeable ice formation (Arnalds, 2015). Moreover, redistribution of snow by wind is commonly observed in the poorly vegetated areas. More details about the monitoring sites can be found in Farbrot et al (2007).

## 3. Methods

### 3.1. Climate data

Long-term climate data are available from the Norwegian Meteorological Institute (MET Norway), either as in-situ observations from nearby weather stations or from high-resolution gridded (1km grid spacing) daily series available as "*seNorge*" data (Lussana et al., 2018a;Lussana et al., 2018b;Saloranta, 2016). For all borehole sites in Norway, we used the daily *seNorge* air temperature, snow depth (SD), precipitation and snow water equivalent (SWE). The elevation of the *seNorge* cell is not exactly the same as the borehole elevation, and strong winter air temperature inversions may additionally bias the *seNorge* data (Lussana et al., 2018b). For some borehole sites, we therefore performed a statistical downscaling, by determining monthly regression estimates between the *seNorge* time series and air temperature measurements at the sites since the installation of the boreholes (max. 10 years). We then used these regressions to estimate daily air temperatures back to 1957.

Similar gridded data sets of air temperatures exist for Iceland, provided by the Icelandic Meteorological Office (IMO), which are for 1-km$^2$ resolution, based on lapse rate adjustment and interpolation between the meteorological stations (Crochet and Jóhannesson, 2011). Snow depth was modelled using a degree-day SWE model (Saloranta, 2012) and HARMONIE gridded precipitation data set (Bengtsson et al., 2017), by the same procedure as for the Norwegian *seNorge* data (Czekirda et al., 2019).

### 3.2. Air and ground surface temperature measurements

At each borehole location, surface air (SAT) and ground surface temperatures (GST) are measured using miniature temperature loggers (MTL) with accuracy and resolution usually better than ±0.2°C. At the Iškoras site, 7 stations measuring SAT and GST were established along a profile line from north to south (Figure 1d), addressing winter temperature inversion





conditions. Shorter data gaps in SAT were filled by neighbouring stations using simple
regression, with $R^2>0.75$.

### 3.3. Ground temperatures

The boreholes at all sites were established during the period 2007 to 2009 (Table 1), besides
Juvvass-PACE which was established in 1999. They are equipped with thermistors coupled to
a logging device, with measurement accuracies between ±0.01 and ±0.2°C (Table 1). The
boreholes at Iškoras and Tronfjell are equipped with PT1000 thermistor strings, measuring
temperature with accuracies better than ±0.01 °C. The data are logged using Campbell logging
devices. The borehole in Guolasjávri is 30 m deep, but the logger chain is only 15 m
(Geoprecision system with Dallas thermistors, ±0.1°C).   A similar system is used at
Lávkavággi, Jetta and the Juvflye observatory. In Iceland, logger systems have been changed
during the monitoring period. At present three boreholes are equipped with Geoprecision
logging systems.

### 3.4. Electrical resistivity tomography (ERT)

ERT yields the 2- or 3-dimensional electrical resistivity distribution of the subsurface by
injecting an electric current between two electrodes coupled to the ground surface and
measuring the resulting electrical potential differences at two further electrodes along a profile
line. By using different combinations of this 4-electrode measurement (so-called quadrupoles)
with various spacings between the electrodes, a 2-dimensional resistivity section can be
obtained. The investigation depth depends mainly on the distances between the electrodes
employed along the profile and the profile length, with larger distance giving greater penetration
depth. The obtained apparent resistivity measurements have to be inverted using suitable
inversion algorithms yielding the specific electrical resistivity distribution along the 2D
profiles. High electrical resistivity values can be associated with frozen conditions including
ground ice occurrences or dry blocky layers, whereas low electrical resistivity values points to
(high) liquid water contents and unfrozen conditions (Hauck, 2002). ERT data acquisition was
conducted with ABEM Terrameters (SAS1000 or LS) using Wenner protocols. All ERT
profiles were inverted using common inversion parameters within the software Res2Dinv (Loke
and Barker, 1995). The length of the profiles varied between 80 and 160 m, and a 2-m-spacing
protocol was used. The repeated ERT measurements were performed in the immediate vicinity


of the borehole locations on Iškoras, Guolasjávri, Juvflye and Tronfjell, with the first measurements in 2009. Measurements were normally carried out at the end of August or early September (Table 1).

### 3.5. Heat flow modelling

For selected sites the ground thermal regime was modelled with the simple heat conduction model CryoGRID2 (Westermann et al., 2013) to reproduce the observed ground temperature evolution, and test the influence of different forcing factors. The subsurface temperature distribution was simulated by numerically solving the transient 1D heat equation (Williams and Smith, 1989). As boundary conditions, we prescribe time series of measured GST for calibration of the subsurface conditions, and the geothermal heat flux at depth (Table 2). For the runs, the snow cover was included using the *seNorge* snow depth data set (Lussana et al., 2018a;Lussana et al., 2018b;Saloranta, 2016), and air temperature from *seNorge* was applied at the upper boundary. This *seNorge* snow data set is not corrected for wind drift, so a wind drift factor is included during the calibration process. The thermal properties of the ground are described in terms of density ($\rho$), thermal conductivity (k) and heat capacity (c). The heat conduction equation was discretized along the borehole depth using finite differences and subsequently solved by applying the method of lines. For details of CryoGRID2, see Westermann et al (2013) and Czekirda et al. (2019) who applied CryoGRID2 spatially for southern Norway and Iceland, respectively.

## 4. Results

### 4.1. Regional climate trends

In northern Europe and particularly in Norway, surface air temperature (SAT) had a positive decadal trend between +0.2 and +0.6 °C dec$^{-1}$ between 1991 and 2020 (Figure 2a). Since c. 1990 we observe mainly higher SAT (between +0.5 and +1.5°C) than average during the current normal period (1991-2020) for all permafrost observatories included in this study (Figure 2b). Northern Norway has the largest positive deviation from the normal, while Iceland has the lowest, with deviations normally below +1°C. There is a trend to increased snow cover, especially in eastern Norway (Tronfjell) and northern Norway (Iškoras and Guolasjávri) (Figure 2c). In central and western Norway (Jotunheimen) the SWE increase was less pronounced (Figure 2c).

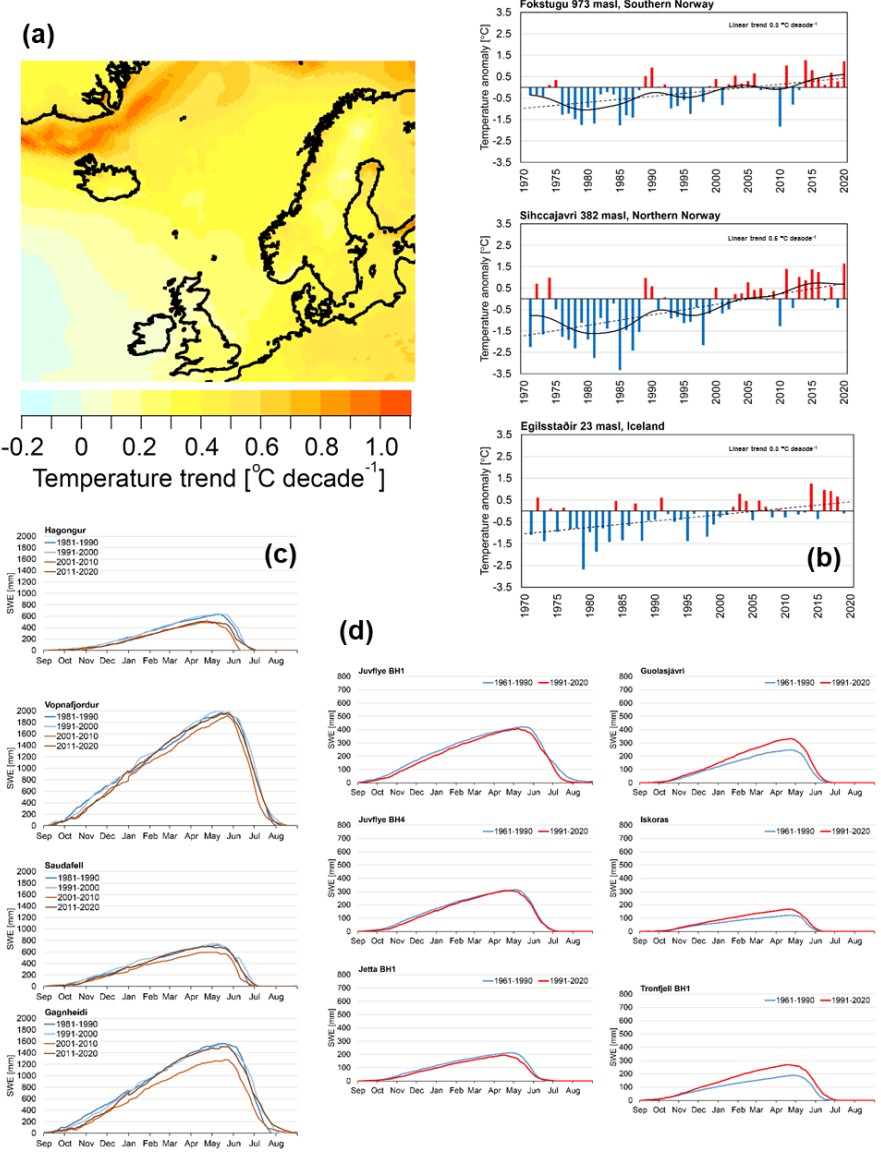

**Figure 2 (a): Decadal air temperature trend during the 30-year normal period 1991–2020 based**
**on ERA5 reanalysis data (Hersbach et al., 2020). (b) Time series of MAAT from 1971 to 2020**
**obtained from official weather stations located near the borehole observatories. Annual values are**
**shown as temperature anomalies with respect to the 1991–2020 average. Gaussian filter (black**
**line) showing decadal variations and linear trend (dotted line) applied, showing the long-term**
**trend. (c) Decadal mean of snow water equivalent (SWE) for Icelandic sites computed using a**
**degree-day SWE model and the Harmonie precipitation data set (Bengtsson et al., 2017). For**
**Icelandic sites the data are calculated for the closest 1 km2 grid cell and a precipitation fraction**
**of 1 and (d) decadal mean of SWE for Norwegian sites obtained from *seNorge* (Saloranta, 2012).**
**For Norwegian sites the data are calculated from nearby grid point with representative height (+/-**
**50 m elevation).**






On Iceland, snow depth is normally much higher than at the Norwegian sites, with slightly increasing trends especially after 2010 in eastern Iceland (Gunnarsson et al., 2019). In central Iceland (Hágöngur), snow cover seems to decrease slightly after 2010 according to our estimations (Figure 2c).


### 4.2. Air (SAT), ground surface temperature (GST) and surface offset (SO)

The surface offset (SO) is defined as the temperature difference between GST and SAT (eg. Smith and Riseborough, 2002), and normally related to snow cover (winter) and vegetation (summer). The average winter offset (GST minus SAT) is positive at all sites, indicating a

higher GST than SAT due to the insulating snow cover (Figure A1). However, the magnitude of the winter offset is different, with the sites at Iškoras, Tron, Jetta and Vopnafjörður on Iceland having average offsets close to +3°C or above (Figure A1). Summer offsets also indicate in general higher GST than SAT, except for the Iškoras site. This may be related to vegetation cover, which cools the ground surface, and/or a more persistent snow cover during spring, when

SAT becomes positive.

At the Norwegian sites, the increase in GST is apparently higher than SAT, while at the Icelandic borehole sites the opposite seems to prevail (Table 1, Figure A2). We observe also a general increase in SO during the measurement period, with trends varying between <+0.5°C dec$^{-1}$ and +1.6 °C dec$^{-1}$. While the average annual SAT has normally been below 0°C during

the measurement period, GST values over time reach more often >0°C. This is especially the case for the sites Jetta, Tronfjell and Iškoras in Norway and Hágöngur in Iceland, facilitating thawing and degradation of permafrost at these sites (see Figure A2).

### 4.3. Ground temperatures (GT)

In general, ground temperatures (GT) at 10 m depth increased during the measurement period (Figure 3, Table 1), although three cold years in 2010-12 led to a temporary cooling of ground temperatures in southern Norway (Figure 3b). Since then, GT increased in an accelerated pace, and the GT trend at c. 10 m depth varied between 0 and +0.5°C dec$^{-1}$ (Table 1). In northern Norway, a warming trend prevailed during the entire measurement period, with values between

+0.4 and +0.5°C dec$^{-1}$ at 10 m depth for all sites. In Iceland, GT trends were also mainly



positive, but below +0.3°C dec$^{-1}$ (Table 1). In general, the warmest years has been recorded since 2018 at all sites with the exception of 2021 and 2022 (Figures 3 and 4). The fastest increase of GT after the cool period in 2010-2012 was observed in southern Norway at sites close to the lower permafrost limit and with large active layer thickness (ALT) (Tron, Jetta,

Juvflye BHs3 and 4), while colder sites with higher ice contents reacted slower (Juvflye BH1 and PACE).

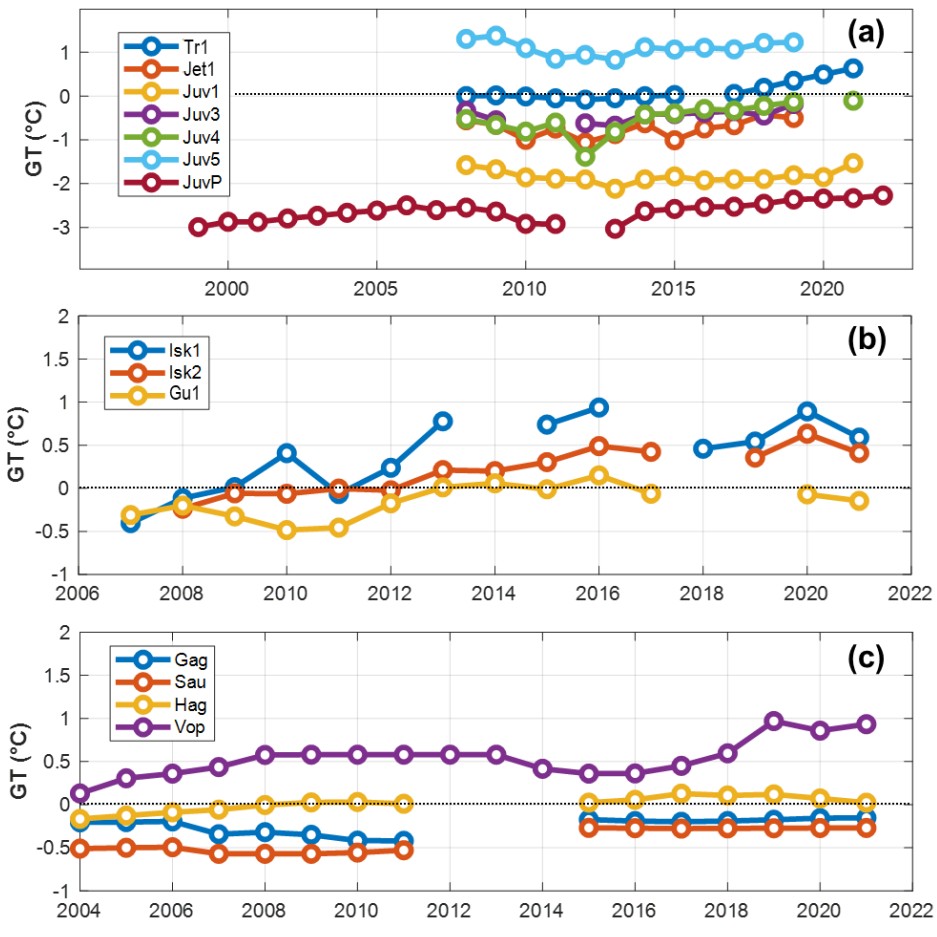

**Figures 3: Ground temperature (GT) development in time at 10 m depth at selected sites calculated over a hydrological year in (a) Southern Norway, (b) Northern Norway and (c) Iceland.**
**At Vopnafjörður in Iceland, GT = 20 m.**






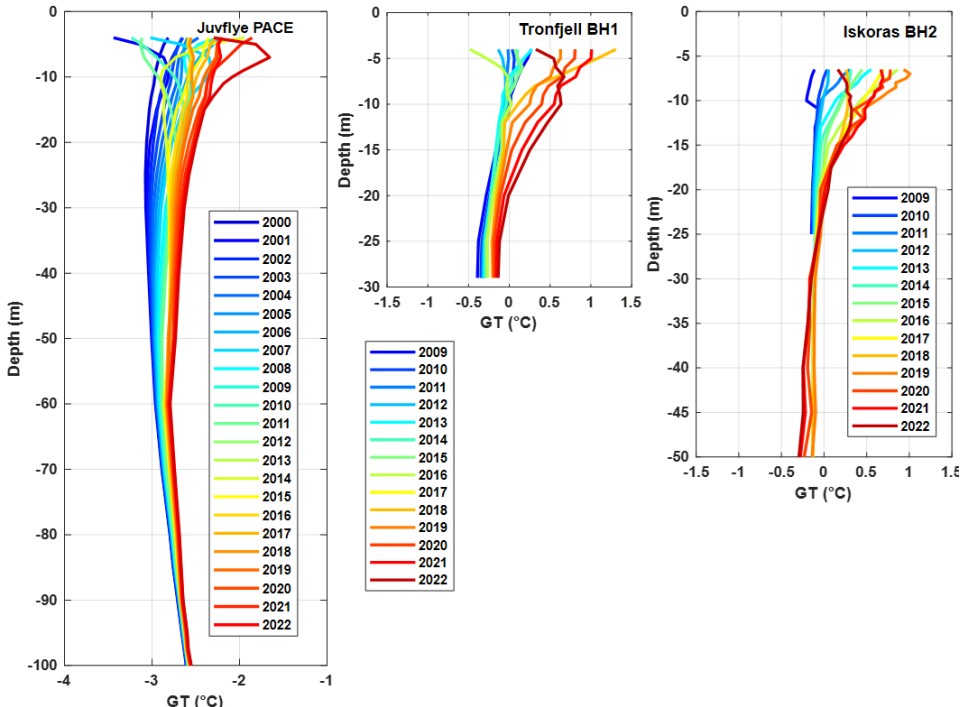

**Figure 4: Annual average GT with depth for Juvflye, Tronfjell and Iškoras, respectively, over the**
**measurement period. The last years were the warmest over the entire observation period and at**
**all depths.**

### 4.4. Active Layer Thickness (ALT)

The ALT development in southern Norway shows a cyclic development because of the cool
period between 2010 and 2012 (Figure 5). However, already one year after the cool years the
ALT at all sites reached the same depth range as in the years before the cool period. The
reduction of the active layer in the 2012/13 season is observed at all sites in southern Norway,
with the most pronounced change at Tronfjell, and the least pronounced in the Juvflye area.
Juvflye BH1 is drilled in a silt-rich cryoturbated moraine above bedrock, and the sediment cover
is more ice-rich, damping the ALT changes. In Northern Norway, ALT has continuously





increased throughout the monitoring period, while in Iceland the main increase was registered after 2015 (Figure B2).

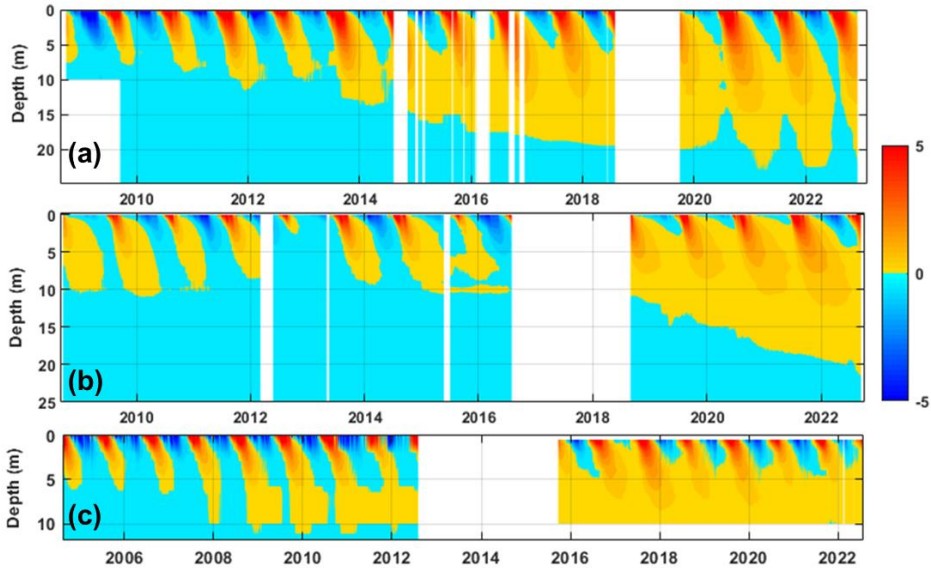


**Figure 5: Time-depth-temperature plot for selected sites at the permafrost observatories. (a) Iškoras BH2. Here a talik developed after 2014, however, the borehole partly re-froze after 2022. (b) Tronfjell BH1. A talik developed rapidly after the data gap between 2016 and 2018. (c) Hágöngur. Here, a talik already has been established since 2012**


At three of our sites in Norway and Iceland a clear talik development could be observed (Figure 5). At Iškoras BH2 a talik started to develop during the winter 2014/2015, following a series of three years with high SAT. This talik evolved rapidly and thawed down to 22 m in 2022, however, the winters in 2021 and 2022 were cool and reversed some of the development (Figure

5a). At Iškoras BH1, which is drilled in pure bedrock, permafrost was not observed within the borehole (10 m), even though the borehole froze back completely at the start of the monitoring period. Also here, a strong warming is observed during the entire monitoring period, with no re-freezing of the borehole since 2014 (Figure B2b). At Guolasjávri we can see a similar development, with thaw deeper than 15 m after 2015, and manual measurements with a

thermistor string indicating positive ground temperatures at 22 m depth in 2019. Until 2020, seasonal freezing down to 15 m was observed, but since then temperatures above 0°C have been registered at 15 m depth (Figure B2b).



In southern Norway, Tronfjell has developed a talik sometime after 2017 (data gap), and at present experiences thaw down to 20 m in 2022 (Figure 5b). After a very cool winter 2012/13

and subsequent cool summer 2013, the ALT at this site was drastically reduced by c. 8 m compared to the years before. After this event, ALT quickly rebounded to similar values as before, followed by an increase in ALT. In the last years, there are signs that the ground does not fully freeze back anymore.

In Iceland, ALT has increased after 2012. A talik developed in Hágöngur already after 2010,

and the borehole is free of permafrost today (Figure 5c). However, at greater depth permafrost may still prevail. At Gagnhaiði, a shallow zone between 4 and 5 meters seems not to re-freeze during winter since 2016 (Figure B2c), however, this measurement must be taken with caution as the measurements can also be related to uncertainties of the thermistor precision (Figure B2c).

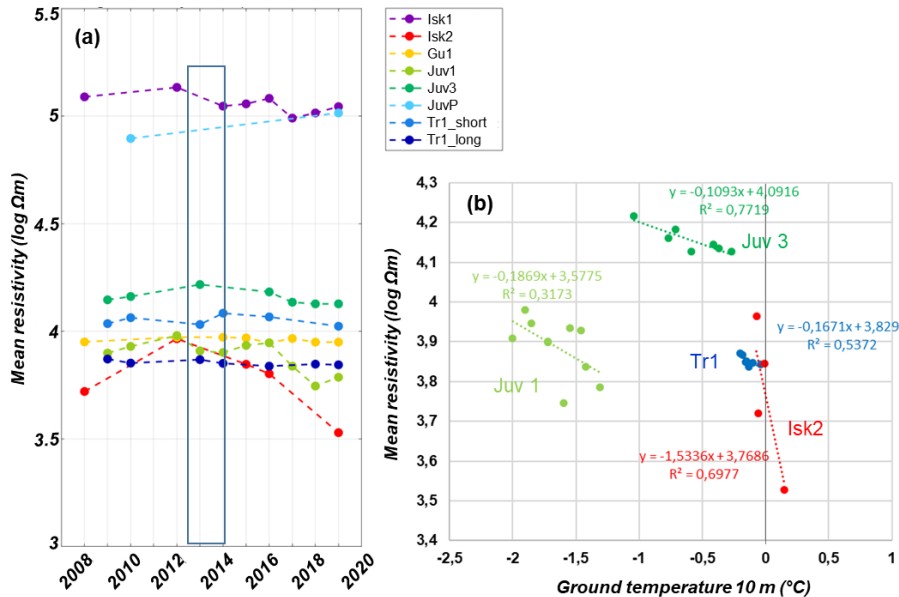


**Figure 6 (a) Development of specific resistivity at borehole locations at the sites where multi-temporal ERT surveys were measured. The values are calculated as a spatial mean over an area (the so-called zone-of-interest, ZOI), mostly 10-20 m wide and a couple of meters deep. This ZOI is considered as a representative permafrost zone below the active layer, at least during the first**

**part of the measurement period. The box indicates the cold period around 2013. (b): Average specific resistivity as in (a), plotted against the average ground temperature at the date of the ERT survey within the same depth range. All sites show a consistent overall decrease of resistivity with increasing ground temperatures, with the most pronounced resistivity change around the melting point**




### 4.5. Geophysical changes

The time series of resistivity changes obtained from the repeated ERT surveys show an explainable pattern for the different profiles (Figure 6a) and can be related to GT variations (Figure 6b). For this, the inverted specific resistivity values were averaged within a so-called

zone-of-interest (ZOI, see Etzelmüller et al., 2020a;Hilbich et al., 2022), which was manually defined around the borehole location and below the active layer depth for each site/profile. In Figure 6b, this mean resistivity value is then plotted against the mean borehole temperature over the same depth range at the date of the ERT measurement. In southern Norway, resistivity values increase slightly during the cool period before 2013 and decrease afterwards. In northern

Norway a stable (Guolasjávri) or decreasing trend (Iškoras) was observed. When relating average resistivity with average borehole temperatures a negative relationship dominates (Figure 6b), as expected from theory (e.g.Oldenborger and and LeBlanc, 2018), varying between -1.5 $\log\Omega m\ °C^{-1}$ at Iškoras to -0.1 $\log\Omega m\ °C^{-1}$ at JuvBH3.

### 4.6. Heat flow modelling (CryoGRID2)

The numerical modelling successfully reconstructed the development of taliks at or close to the timing of the observations, indicating that most of the thermal patterns in the ground can be explained by conductive heat flow modelling alone (Figure 7a). At Iškoras, the onset of the talik formation could be reproduced exactly, along with the appr. thaw depth. ALT during the

cooler part of the model period before 1990 was around 5 m, increasing to 10 m after 2000. At Tronfjell (Figure 7b) the fit between simulated and observed temperatures was worse, however the latest talik development was reproduced, along with the observed thaw depth. The model implicated large ALT and almost talik formation early in the 2000s, while the observed shallow ALT of below 2 m in 2013 was reproduced. According to the model, ALT was close to 2-3 m

until 2000, where a strong increase of ALT was simulated. This seems related to variations in snow thickness, which had an increasing trend since 2000. This is in accordance with observations of snow height development in Norway (e.g. Dyrrdal et al., 2012).



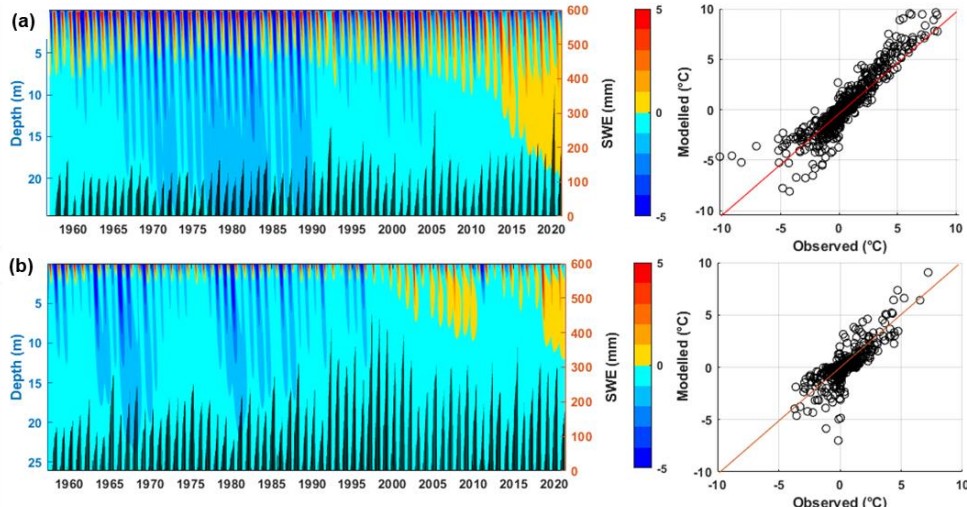

**Figure 7: Results from ground heat flow modelling at two boreholes (a) Iškoras BH1 and (b) Tronfjell BH1 with observed talik development for the period 1957 to 2020. (Left subplots) Modelled time-depth-temperature plot, together with the black bars indicating modelled snow water equivalent (SWE) [mm] at the sites based on (Lussana et al., 2018a;Lussana et al., 2018b;Saloranta, 2016). (Right subplots) Validating scatter plots for all GT between 0 and 10 m depth for the period 2009 and 2020. Both sites show talik development and demonstrate that the last decade was the warmest since 1957. SWE has increased by 50 and 82 mm dec$^{-1}$ for Iškoras and Tron, respectively, during this period.**

## 5. Discussion

### 5.1. Permafrost dynamics

The observed GT developments presented in this study are all in line with recent publications of permafrost dynamics in a changing climate. Permafrost warming and degradation seem to be more rapid in the north than in the south and the maritime west, which is consistent with previous research (Etzelmüller et al., 2020a;Biskaborn et al., 2019;Romanovsky et al., 2010;Christiansen et al., 2010;Smith et al., 2022). Warm permafrost sites normally show slower thermal response than colder sites due to latent heat processes (Romanovsky et al., 2010;Smith et al., 2022), however, at our sites water/ice contents are low, facilitating fast thermal response. Finally, the highest permafrost temperatures were recorded between 2019 and 2021 at all sites (Etzelmüller et al., 2020a).

Trends in GTs are consistent with trends in SAT. The 2011-2020 decade was the warmest on the SAT record in Norway and Iceland and most of the years 2014 through 2022 rank among the warmest years on record (updated time series from MET Norway and IMO). Talik



development was observed during the last part of the monitoring period in all permafrost observatories. Such drastic ground temperature development is normally due to an increase in GST, either due to higher SAT or a change of snow cover and composition.

In addition, there is a clear tendency to increasing snow depth during the monitoring period, along with a shortening of snow cover duration with both later snow onset and earlier snow disappearance (Etzelmüller et al., 2020a). The later snow onset seems not to be accompanied by more freezing of the ground, but an increased thawing degree-days (TDD) during fall (Figure C1b). It was also speculated that more frequent and intense rain-on-snow (ROS) events (Pall et

al., 2019;Westermann et al., 2011;Rizzi et al., 2018) and winter warm spells form ice layers near the snow surface, thus reducing snow surface erosion due to wind and leading to a thicker winter snow cover. There are no clear observations of this phenomenon, however there are clear observations of more rain on snow events in Norwegian mountains, influencing snow composition, thickness and thermal conductivity (Rizzi et al., 2018;Dyrrdal et al.,

2012;Vikhamar-Schuler et al., 2016). Our numerical modelling indicates that the variations of SAT and snow depth from *seNorge* (Lussana et al., 2018b) alone could predict the onset of the talik reasonably well. Furthermore, thermal preconditioning is discussed, e.g. heat waves reducing the ice content in the ground and thus conditioning the ground to develop taliks more easily. This could be the case at Tron mountain where a smaller talik was modelled just after

2000 (Figure 7). This was also discussed in Isaksen et al. (2011), which observed first signs of talik formation on a permafrost monitoring site on Dovrefjell between 2006-2009, and formation of a talik in a model for the same three years (2006-2009) at Juv-BH5, which today has no permafrost in the upper 10 meters. However, this process is not part of the simple heat flow modelling scheme.


### 5.2. The influence of ground characteristics

Except the boreholes at Juvflye (BH1) and Trond (BH1) the boreholes are drilled in coarse sediment cover or in bedrock with only a thin sediment cover of less than 2-3 m (Farbrot et al., 2007;Farbrot et al., 2011;Farbrot et al., 2013) and relatively small ice content. Permafrost in

Scandinavia is mostly restricted to mountain environments, besides the Finnmarkvidda area, where permafrost is widely encountered in palsa mires and peat plateaus (Borge et al., 2017;Martin et al., 2021;Kjellman et al., 2018). In the mountains, thin sediment thickness above bedrock dominates with few exceptions. This makes mountain areas fast to respond in comparison to the more ice-rich arctic areas, especially if ALT exceeds the general sediment



thickness. Thus, the response of near-surface ground temperatures (c. < 20 m) to changing climate forcing is fast to immediate. At the Iškoras site we observe a partial reversal of the degradation development (Figure 5a). This indicates very low water content in the bedrock and the very high thermal conductivity of the underlying quartzite, with values measured in bedrock cores from the site of >5 W m$^{-1}$ °K$^{-1}$ (Farbrot et al., 2013).

This is also confirmed by the ERT trend between resistivity and average ground temperature varied between -1.5 logΩm °C$^{-1}$ at Iškoras to -0.1 logΩm °C$^{-1}$ at JuvBH3. The large trend at Iškoras is reflecting the (strong) decrease of resistivity upon thawing close to the melting point, where the liquid water content strongly increases and the mobility of the ions in the pore fluid increases as well. The large variation of the gradients in the negative temperature range can be

related to bedrock type and moisture/ice contents. The smaller gradient at JuvBH3 is related to a small moisture/ice content, the larger gradient at JuvBH1 corresponds to an increased ice content (cf. Hauck, 2002).

### 5.3. The influence of air temperature inversions

Winter air temperature inversions and change of inversion patterns will highly influence the thermal regime at local sites. Normally, the frequency and magnitude of winter inversions increase with continentality (Figure D1a). In extreme cases, valley bottom temperatures can become much lower than higher up in the mountains, even in an annual average, as observed e.g. in continental mountains sites in Yukon and Alaska (Lewkowicz et al., 2011;Lewkowicz

and Bonnaventure, 2011). This climate pattern might lead to the preservation of palsas and peat plateaus in the valley bottom, while the nearby mountain peaks at higher elevations may experience degrading permafrost. This inversion pattern is also visible in eastern Norway (Tronfjell), although less extreme, while all other areas may have occasional inversions during winter, but with overall negative monthly lapse rates (Figure D1b). The frequency and

magnitude of inversions is likely influenced by climate change, and permafrost in different altitudinal zones may thus react differently to the same large-scale changes. The permafrost observatories in Norway are all located close to the mountain tops, while the valleys and even the lower parts of the slopes are generally permafrost-free. It is therefore likely that the ground temperature trends presented in this study are largely representative for the mountain permafrost

domain in Norway and Iceland. However, permafrost in lowland areas, especially in palsa mires in Finnmark, may potentially experience different trends in SAT due to changes in inversion patterns. Furthermore, we emphasize that transferring SAT trends measured in valley settings





to higher elevations may lead to strong biases when assessing the impact of climate change on mountain permafrost.


## 6. Conclusions

Based on direct temperature measurements in permafrost boreholes in Norway and Iceland between 2004 (1999 at Juvvasshøe) and 2022, as well as repeated electrical resistivity tomography and long-term permafrost modelling the following conclusions can be drawn:

- Permafrost in Norway and Iceland is warming at a high rate.
- On several sites, development of taliks or complete permafrost degradation is observed.
- At most sites GST is apparently increasing stronger than SAT. Changing snow conditions appear to be the most important factor for the higher GST rates.

The observation record clearly demonstrates the impact of climate change on the thermal state
of permafrost in Norway and Iceland.

Several of the Norwegian sites will be continued as part of the national operational permafrost monitoring program (Isaksen et al., 2022) and become available in near-real time on https://cryo.met.no/en/permafrost.





**APPENDIX A - Surface offset (SO)**

Surface offset (SO) is the difference of SAT and GST and highly influenced by snow and vegetation cover. Figs. A1 and A2 are both related to SO and show the relative influence of especially snow cover (vegetation cover is low at all sites) in space (at borehole sites) and time.

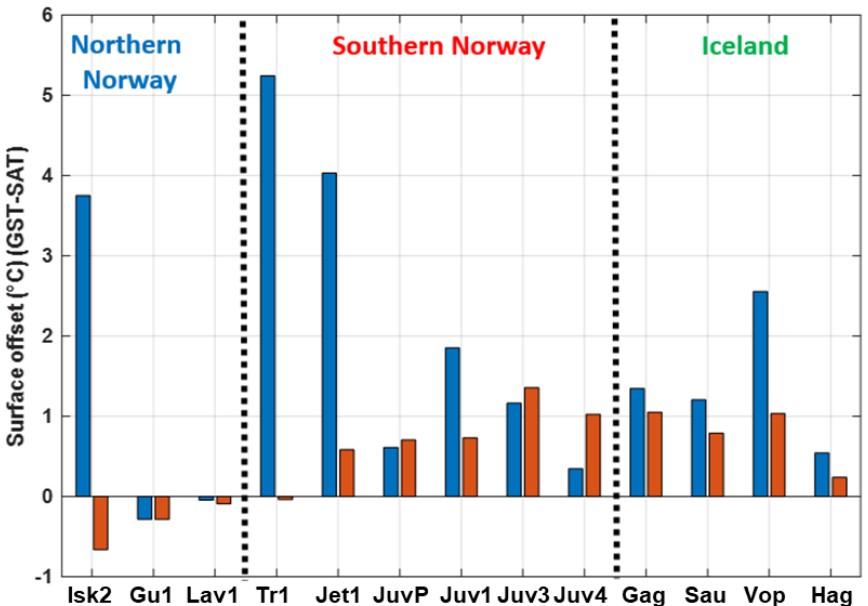


**Figure A1: Average surface offset (GST-SAT) for selected boreholes in Norway and Iceland. Winter (blue) and summer (red). Most sites show positive winter and summer offsets, indicating warmer conditions at the ground surface than in the air.**








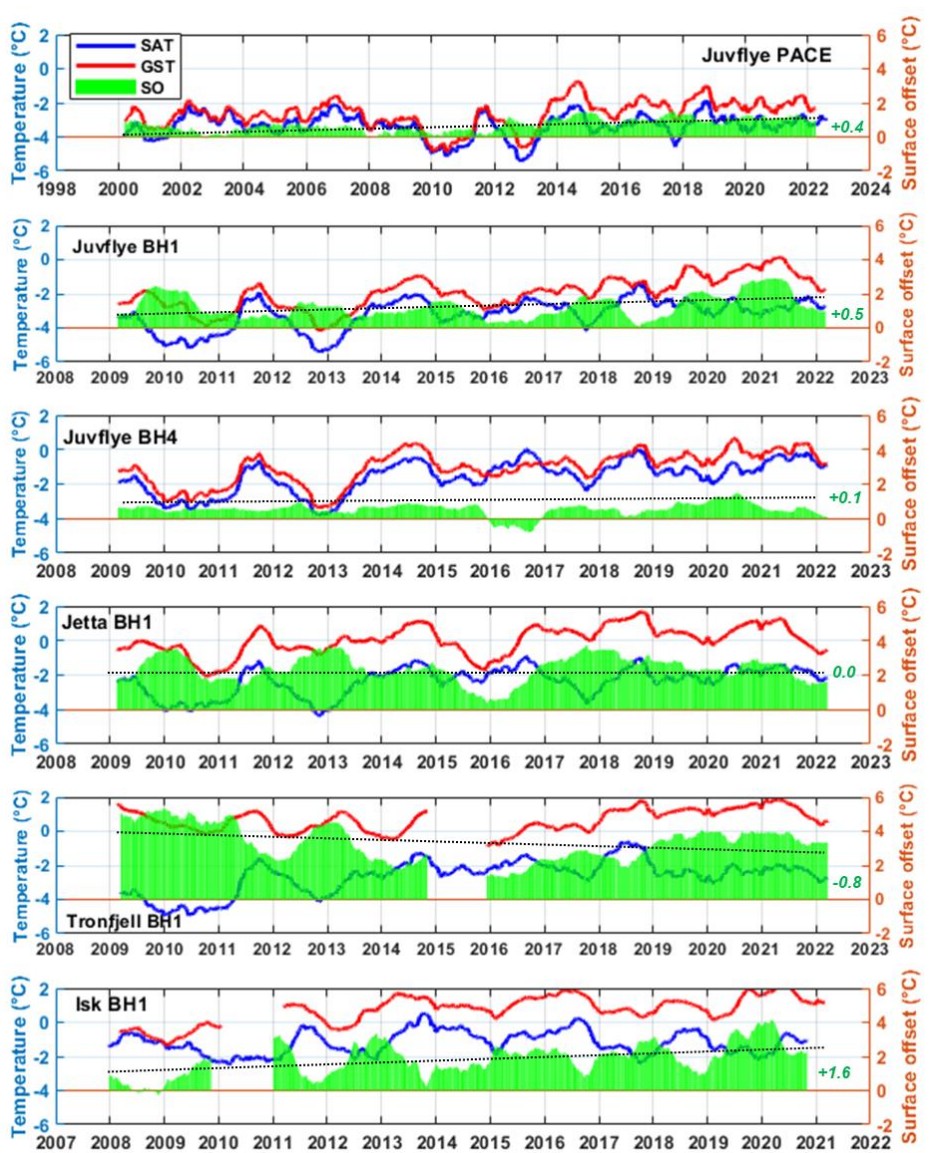

**Figure A2: Average daily SAT, GST and SO development at selected boreholes in Norway and Iceland. The curves show a 365-days moving average based on a Gaussian filter. The trend lines**
**denote the SO trend, while the green numbers denotes the trend of SO in °C dec⁻¹. The trend varies between 0 °C dec⁻¹ for Jetta BH1 and +1.6 °C dec⁻¹ for IškorasBH2. Tronfjell has a negative trend with -0.8 °C dec⁻¹, probably related to the transition from mainly negative GST in the start of the period towards positive GST.**






## APPENDIX B - Ground temperatures and active layer thickness

The following graphs show the development of ground temperatures and ALT for all borehole sites. For Figure B1 the ALT is defined as the largest depth for the 0°C contour during the hydrological year. The deviations in percentages are related to the average ALT during the measurement period.

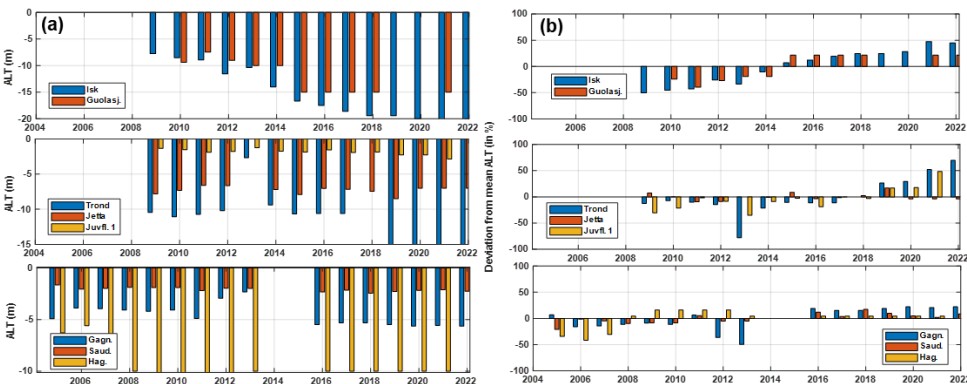

**Figure B1: (a) Active layer thickness development at selected boreholes at the permafrost observatories. ALT exactly at -15 m or -10 m denotes thaw in the entire borehole length and normally talik development (Figure 5). (b): Normalised active layer thickness change in relation of overall average during the measurement period in percent. In northern Norway a steady increasing trend is observed, while in southern Norway changes were less pronounced and also negative during a couple of years around 2013.**



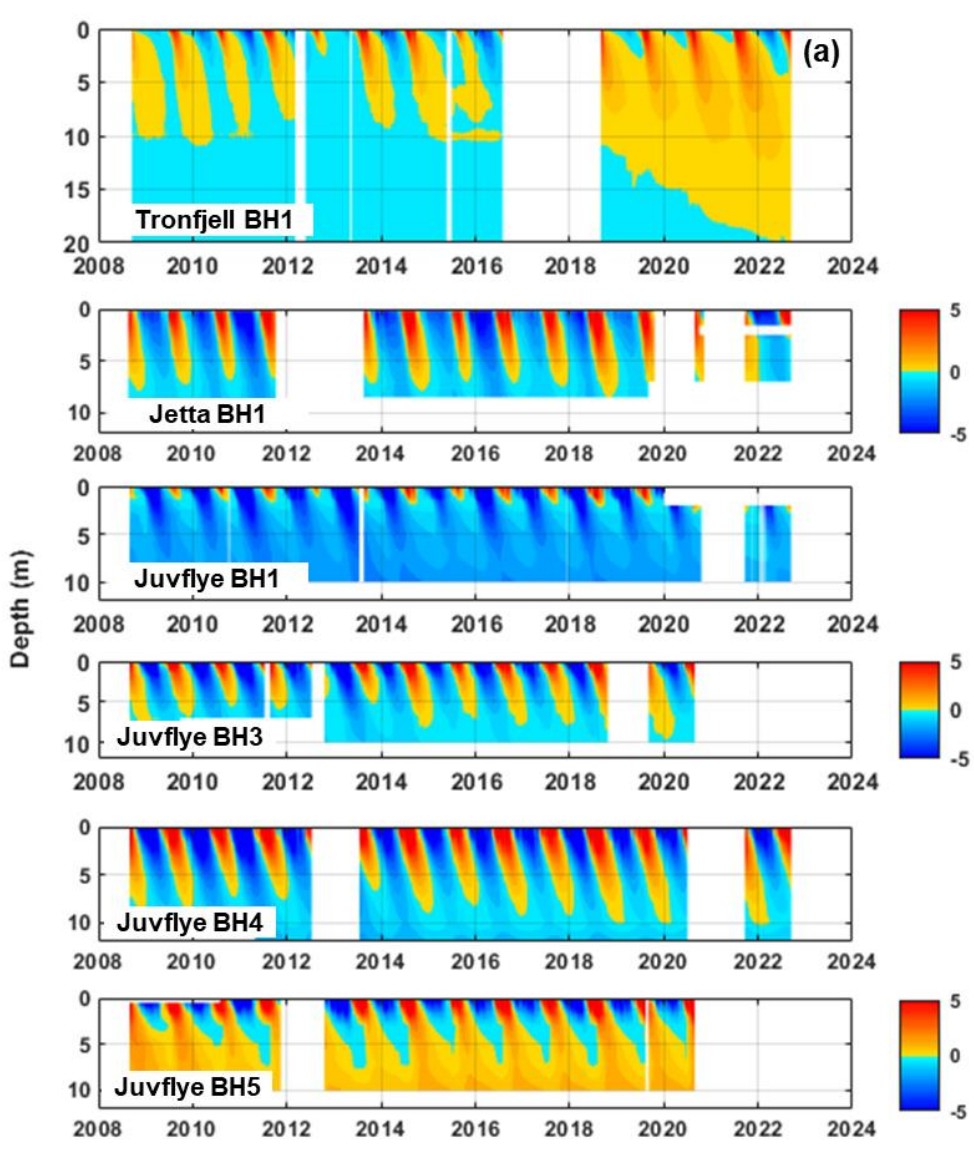

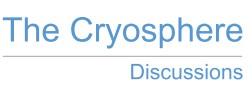
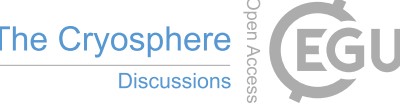

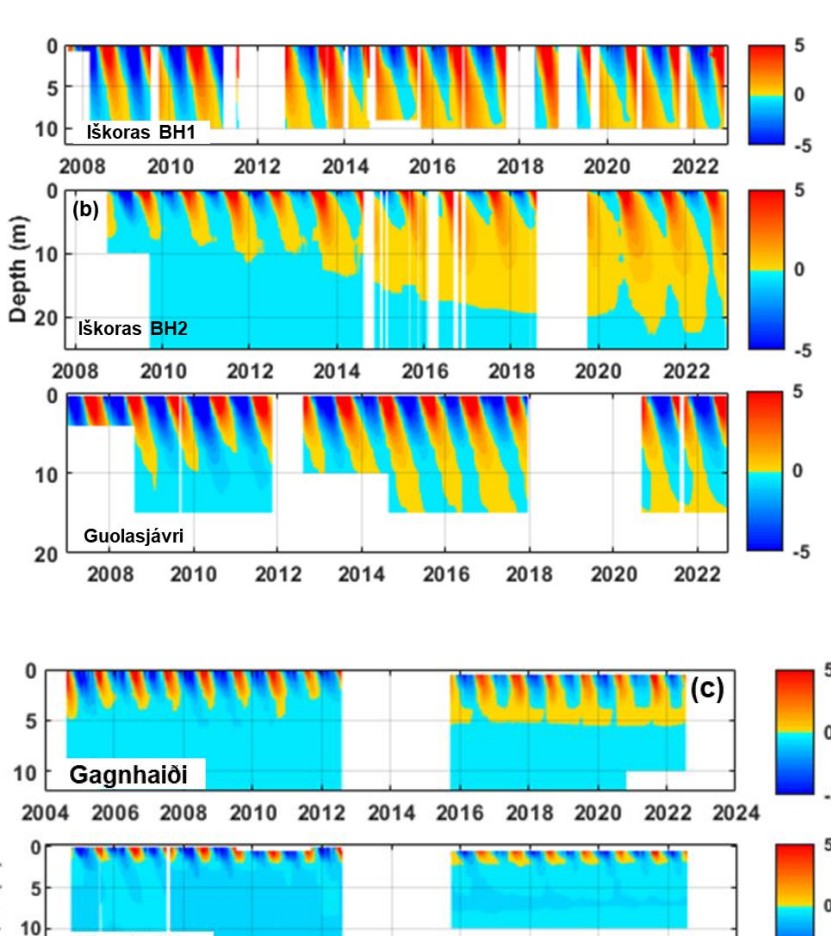

**Figure B2: Time-depth temperature plots for all measurement sites. (a): Sites in southern Norway, (b) Northern Norway and (c) Iceland.**





<sub>565</sub> **APPENDIX C - Seasonal variations of ground surface temperatures (GST)**

Seasonal variations of GST display changes of the energy forcing conditions on top of the ground surface and below snow and vegetation cover. There is in general limited positive trends for summer thawing degree days, while winter freezing degree days are highly depending on
<sub>570</sub> snow cover and increasing for most sites in varying pace (e.g. Juv-PACE). Thawing degree days during the shoulder seasons seems slightly increasing for spring, with a strong increase during fall.

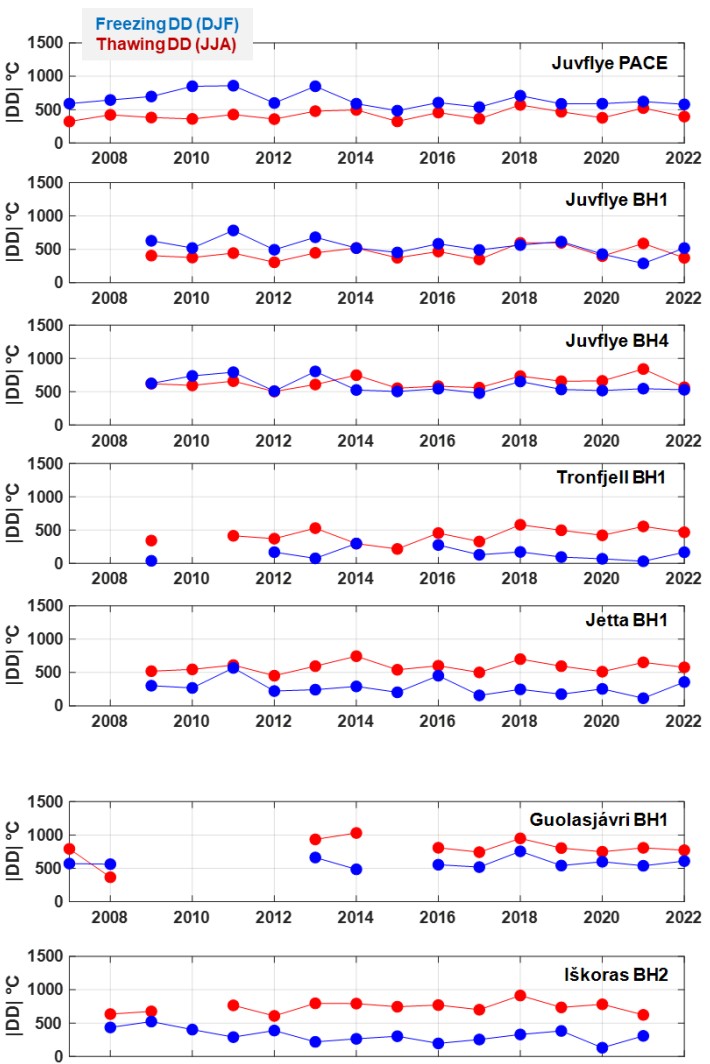





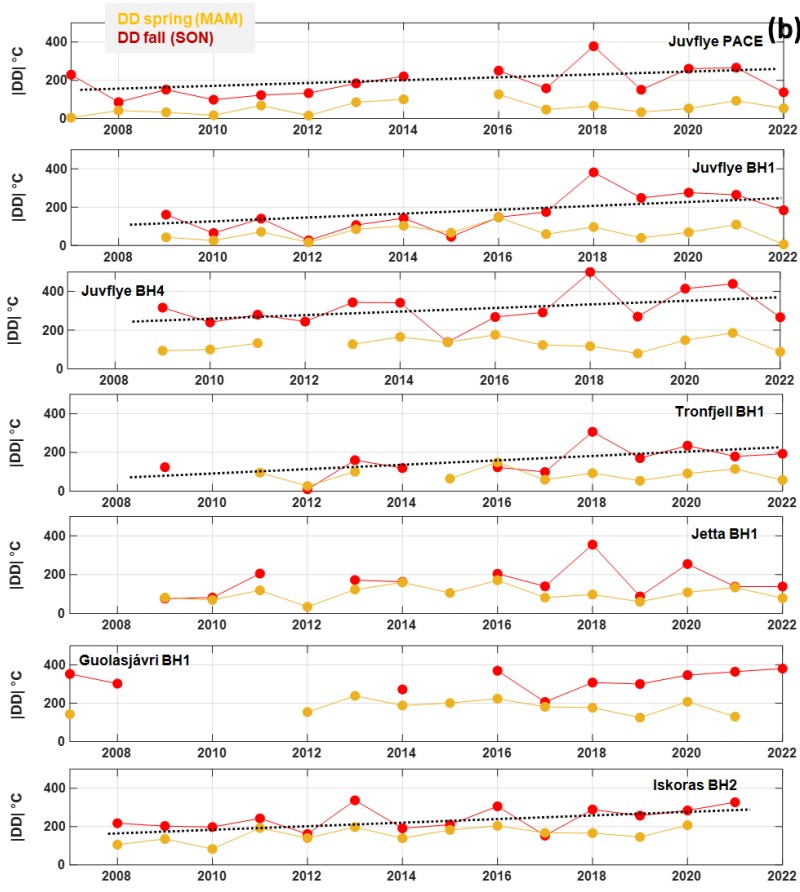

**Figure C1: Seasonal degree day (DD) development of GST during the measurement period. (a) freezing and thawing DD during winter and summer, respectively. Winter = DJF (December, January February), Summer = JJA (June, July, August). All sites show a trend of winter DD decrease and summer DD increase, respectively. However, winter DD decrease was higher (+50-100 DD°C dec⁻¹) than summer decrease (-10 - -50 DD°C dec⁻¹). (b) DD during the shoulder seasons for spring = MAM (March, April, May) and fall = SON (September, October, November). All sites show a positive trend towards higher DD, however, the trend during fall is much higher with values between 60-150 DD°C dec⁻¹ in relation to spring values (<10 DD°C dec⁻¹).**
## APPENDIX D - Inversion settings at the study site

In Norway, inversions are frequent in the Finnmark area (Iškoras) (Figure D1a) and in the
eastern parts of southern Norway (Tronfjell) (Figure D1b). In Iškoras, we observe strong witner
inversions between the valley bottom and the tree limit, and "normal" negative lapse rates above
(Figure D1c). During the winter months DJF, the average monthly air temperature in the valley
bottom is colder than on the mountain top producing positive lapse rates. During spring and
fall, lapse rates are close to 0°C/100m, while during summer lapse rates of c. -0.5°C/100m are
common (Figure D1c). Towards the coast, normal negative lapse rates dominate, with values
around -0.5°C/100m at our borehole locations. In southern Norway, Tronfjell shows a similar
pattern as Iškoras (Figure D1d). The magnitude of the inversion during the winter months is,
however, less pronounced than in Finnmark (Figure D1a,b). Further west towards the Juvflye
permafrost observatory the inversion pattern is visible during the winter months, but far less
pronounced (Figure D1e).

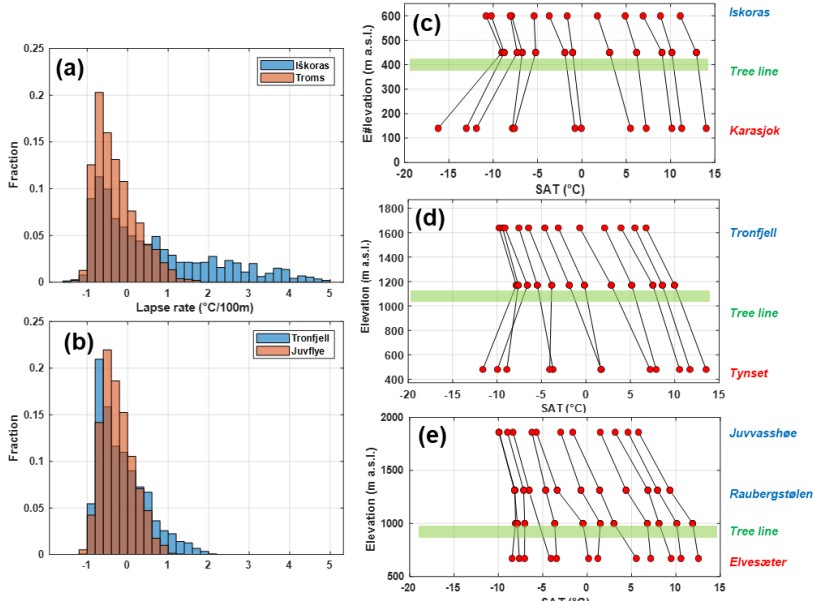

**Figure D1: Frequency and magnitude distribution of daily lapse rates for the permafrost
observatories, calculated for the winter months (DJFM) based on SAT observations. (a) Northern
Norway - Iškoras: Between BH2 meteorological station and Karasjok (500 m elevation difference).**
**Troms: Between Nordnes climate station (c. 600 m a.s.l.) and Skibotn. (b) Southern Norway -
Tronfjell: Between BH1 and Tynset climate station (1100 m difference) and Juvflye: Between
Juvvasshøe and Elveseter climate station (1200 m difference). The blue bars show lapse rate
frequencies for the more continental sites. Seasonal lapse rates for the Iskôras (c), Tronfjell (d)
and Juvflye (e) observatories. The green areas denote the forest. The lapse rates are based on the**
**SAT-GST stations along elevation gradients showed in Figure 1.**



**Author contribution:** BE has initiated and followed up this study, analysed the data and wrote the first drafts of the manuscript. BE has leaded the projects which established the boreholes in southern Norway (except Juvvlye-PACE) and Iceland, and participated in the projects responsible for the reminder boreholes. KI contributed with the data from Juvflye-PACE and Iskoras-BH2, along with the analyses of climate development. JC helped with the modelling exercise and provided the snow data from Iceland. SW helped with the modelling CryoGrid2) and both CHa and Chi provides and analysed the ERT information. All authors contributed to the writing and revision of the manuscript.

**Competing interests:** Cha and KI are members of the editorial board of the "The Cryosphere". The peer-review process was guided by an independent editor, and the authors have also no other competing interests to declare.

**Acknowledgements –** The data collection was carried out during many years, and mainly made happen by help through the academic institutions of the main authors, such as the Norwegian Meteorological Institute and the Universities of Oslo, Fribourg and Zurich in Norway and Switzerland, respectively. The data collection was aided by the help of many individuals during several years of field work, scientific discussion and cooperation; we therefore want to thank sincerely (in alphabetical order): Martin Bathen, Hanne H. Christiansen, Trond Eiken, Herman Farbrot, Regula Frauenfelder, Kjersti Gisnås, Tobias Hipp, Ole Humlum, Cécile Pellet, Siri Jakobsen, Karsten Vedel Johansen, Antoni G. Lewkowicz, Karianne S. Lilleøren, Benjamin Mewes, Thomas V. Schuler, Rune Strand Ødegård. We want to thank all mentioned institutions and individuals.

**Financial support -** This study is based on results and implementation of scientific equipment from the funding of various research projects. The deep borehole *Juvflye-PACE* was drilled during the EU 7th Framework program (PACE - Permafrost and Climate in Europe; ENV4-CT97-0492 and BBW 97.0054-1), and later supported by The European Science Foundation (PACE21; NW.GC/24 Network 112). The Norwegian Research Council funded the boreholes in northern Norway via the IPY-TSP Norway project (grant no.: 176033/S30), in Iceland ("Permafrost on Iceland", grant no. 157837/V30), and in southern Norway (CRYOLINK – "Permafrost and seasonal frost in southern Norway", grant no, 185987/V30). Permafrost Young Researchers Network's contribution to the TSP project in the Nordic



countries. The Swiss National Science Foundation (project TEMPS, CRSII2 136279), the
German National Science Foundation SPCC grant HA3475/3-1) supported in addition the
geophysical surveys.

**Review statement:** This paper was edited by NN and reviewed by NN, NN and NN.

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
