# Peer review of "Rapid warming and degradation of mountain permafrost in Norway and Iceland"

_The Cryosphere, 2023_

## Referee Comment (RC1)

Review of the paper:

**Rapid warming and degradation of mountain permafrost in Norway and Iceland**

By Bernd Etzelmüller et al.

The paper presents data on permafrost changes since 2004. Ground temperatures have risen quicker than surface temperatures and the paper attempts to describe the cause of this.

Novelty

The paper is within the scope of the Cryosphere and it goes beyond current understanding of permafrost development. It´s strength and novelty are the new and extensive dataset presented. The methods used are not unique but appropriate for the paper.

Quality

It presents new data and data sources from Norway and Iceland. The authors are well aware of the quality of the data and the analyses are sound and important for the scientific community. The purpose of the work is well described and the results are discussed in an appropriate way and they are making use of the references.

Significance

The paper presents important data for the scientific community and for policy makers and it will probably have a substantial impact. It is not really ground breaking but it has a clear and urgent message that permafrost is thawing in an increased speed.

Presentation quality

The paper is well organized and it is clearly written and the illustrations are all very good. There are some minor details that can be adjusted:

The apendix references are a bit unclear. A reference such as (Figure D1b) should rather we written (App. D, fig. 1d)

I am not familiar with the use of negative coordinates. For example, the site Saudafell with the coordinate -15$^o$E. Why not 15$^o$W as it is normally written?

On line 28, there is a reference to van Everdingen 1998 which is missing in the reference list.

In the reference list there is a Smith and Riseborough 2002 paper that I cannot find in the text.

---

## Author Comment (AC1)

TC – Submission of

**"Rapid warming and degradation of mountain permafrost in Norway and Iceland"**

by Etzelmüller et al.

**Comments to the reviewers.**

We thank both the reviewers and the editor for thoughtful comments. In the following the review comments are given in red, while our responses are given in black.

**Review 1 (Pelle Holmlund)**

The paper is well organized and it is clearly written and the illustrations are all very good. There are some minor details that can be adjusted: The appendix references are a bit unclear. A reference such as (Figure D1b) should rather be written (App. D, fig. 1d).

I think we follow the proper instructions given for the journal.

I am not familiar with the use of negative coordinates. For example, the site Saudafell with the coordinate -15oE. Why not 15oW as it is normally written?

Right, we changed

On line 28, there is a reference to van Everdingen 1998 which is missing in the reference list. In the reference list there is a Smith and Riseborough 2002 paper that I cannot find in the text.

Corrected

---

## Author Comment (AC2)

TC – Submission of

**_"Rapid warming and degradation of mountain permafrost in Norway and Iceland"_**

by Etzelmüller et al.

**Comments to the reviewers.**

We thank both the reviewers and the editor for thoughtful comments. In the following the review comments are given in red, while our responses are given in black.

**_Review 2_**

_The paper "Rapid warming and degradation of mountain permafrost in Norway and Iceland" by Etzelmüller et al. presents results of permafrost monitoring through borehole temperature and geoelectrical surveys coupled with heat flow modeling to discuss the main heat transfer processes responsible for the reported observations. (…)_

_However, despite it is concise and well-written, I miss some detailed explanation about the physical processes that could explain specific observations (notably about the snow, vegetation and ice content effects) as well more comparison with recently published data. In the same way, some sections (e.g., site descriptions, modeling approach, conclusions) are very general and lack detailed explanations. Therefore, I recommend to detail a little bit more the MS and take into account a few minor recommendations before accepting the manuscript._

We understand the comment and have tried to explain the observations better and made some adjustment to accommodate the comments, both in terms of site description, modelling parameters and the conclusions.

**_GENERAL COMMENTS_**
_Trends: Trends are given for SAT, GTS, etc., but no information about their calculation is given. Could you explain how the trends are calculated? Are there comparable to other trends recently published in the literature?_

Linear trends are always calculated as normal linear regressions $y=ax + b$ between time and temperatures, and long-term decadal changes are based on the slope of the regression ($a$). We added this into the table and figure texts where necessary. The

trends follow those reported from recent publications, and this is mentioned in the first paragraph of the discussion.

*Section. 2. More details could be given to better discuss the results afterwards. For example, there is no information about the regional limits of permafrost or about the type of permafrost (discontinuous…) for each area. This information is always interesting because it is often inferred from modeling of a steady state and comparison with existing temperature data that represent a transient state give valuable information to assess ground thermal conditions and their disequilibrium with current climate conditions.*

We made some additional comments to the sites in terms of permafrost distribution. In general, discontinuous and sporadic permafrost dominate in the Norwegian and Icelandic mountains, mostly because of the influence of the snow cover. We made some more general statements about the permafrost conditions.

*Furthermore, it is not clear how BH that are presented in the study are chosen and why some are left out. Could you explain BH choice?*

Figure 1 show all boreholes drilled, and some of the are drilled in seasonal frost (no permafrost in the borehole). As this manuscript deals with permafrost warming and possible degradation, we omitted some of the less relevant boreholes. But we mean that it is important to mention and partly show some data also of these sites (e.g., Juvflye-BH5, Lavkavagge in northern Norway and Vopnafjördur on Iceland in Table 1).

Sect. 3.5. Detailed information about the model setting and parameters (spatial resolution, thermal propertied, geothermal heat flux, ground ice content…) are missing. One important factor to reproduce ground temperature evolution is the ice content. How is the ice content set up in the modeling experiment for the different sites? The ice content is sometimes mentioned when presenting the data (e.g. L 410) but it is not explicitly discussed. It may have a primary influence on talik development and I think this would deserve clearer description and discussion.

The CryoGrid-2 code is explained elsewhere (e.g. Westermann et al 2013, Czekirda et al. 2019). We see the lack of some more details, and added a Table 2 for the modelling parameters used for the two sites presented in the manuscript. The water/ice content is prescribed (Table 2) and does not change during the modelling period. Table 2 reads now:

| | Iskoras BH2 (Isk2) | Tronfjell BH1 (Tr1) |
|---|---|---|
| Thermal conductivity of bedrock (W K$^{-1}$m$^{-1}$) | 5.5 | 4 |
| Geothermal heat flux (W m$^{-2}$) | 0.05 | 0,03 |
| Density of snow (kg m$^{-3}$) | 350 | 300 |
| Thermal conductivity of snow (W K$^{-1}$m$^{-1}$) | 0.31 | 0.23 |
| Prescribed ground stratigraphy (m): volumetric water/mineral/organic material content (in %) | < 1.5 m: 10/75/0
1.5-2 m: 20/75/0
> 2 m: 2/98/0 | < 1.5 m: 15/85/0
1.5-3 m: 10/90/0
> 3 m: 3/98/0 |

*Table 2: Model parameters and pre-scribed stratigraphy for the Iskoras and Tronfjell site. For more details on value selection and implementation see Westermann et al (2013).*

We added the sentence:
"The thermal properties of the ground are described in terms of density (ρ), thermal conductivity (k) and fraction of mineral, water/ice, organic material and air."

Conclusions are really short and very general. This would be worth detailing a little bit more the statements: e.g. what is meant by a "high" rate? There is no word about differences between Iceland and Norway while they are discussed in the core of the MS, etc.

We have extended the conclusions a bit. They main points read now as follows:

- *"Permafrost in Norway and Iceland is warming with rates between 0 °C dec-1 and 0.6 °C dec-1 (Isk2) in 10 m depth since the start of the measurements. W*
- *In all regions studied, development of taliks or complete permafrost degradation is observed, such as in Tronfjell (southern Norway) and Iskoras (northern Norway). The talik development could be modelled by heat conduction alone and increasing SAT and snow depth as main forcing variables since 2010.*
- *At most sites ground surface temperature (GST) is apparently increasing stronger than surface air temperature (SAT). Changing snow conditions, especially related to increasing snow depth and a shortening of snow cover duration, appear to be the most important factor for the higher GST rates. A thicker winter snow cover may be related to more frequent and intense rain-on-snow events and winter warm spells, that may reduce snow surface erosion due to wind. Further studies are needed to confirm this hypothesis."*

**DETAILED COMMENTS**

*L 17: Seismic data are mentioned only at this line in the whole MS. Either remove this mention or give more detail about these data in the paper.*

removed

*L 25: What do you mean by « increasing » snow cover? Duration? Height?*

We meant "snow depth."

*L106: In title of 2.3. coordinates are not given while they are given for other areas.*

These are "observatories" covering larger areas with different boreholes. Therefore, the location is given in the paragraph, it is the same for 2.5. (Iceland).

*L137: Title is repeated*

Corrected

*L143-145: This level of information about hydraulic characteristics is only given for this area. Try to homogenize the different sub-sections.*

Ok, we harmonized the information.

*L160-16: Could you provide information about the quality of the regressions? Why regressions based on monthly data to predict daily values? This mismatch has likely an influence on the accuracy of the final results.*

We used daily temperature data for the statistical downscaling, and produced regression coefficients for each month ("monthly regression estimates", not monthly data). This procedure corrects the seNorge-data especially during winter with frequent inversion settings in the more continental regions in Norway (east of the coast line)

*L199ff: what do you mean by "high" or "low" resistivity? Could you provide a range of values?*

We added "relatively" and give a number for resistivity. The problem is that the absolute values often are less relevant than the relative change of values within a profile, depending on site-specific conditions. However, all our profiles are connected to ground temperature measurements. We use here only the multi-temporal ERT results to underline the development of permafrost temperature in time.

*L220: how is this wind drift factor defined and applied?*

We have a factor reducing snow depth due to wind drift, fitting modelled and measured ground surface temperatures. We added a Table 2 (which was referenced to, but not included) into the text, which shows the parameters for the modelling, including assumed stratigraphy, geothermal heat flux and other properties. However, as we used a factor of 1 (no changes) for the modelling of the two particular sites, we removed the reference to a wind drift factor.

L223: What is the method of lines?

The method of lines is a numerical technique for solving partial differential equations (i.e., differential equation with at least two independent variables), such as the heat diffusion equation, where the original equation is approximated using ordinary differential equations (i.e. differential equation with only one independent variable). The latter are easier to solve than the original partial differential equation. More details about how it is implemented in detail can be found in Westermann et al. (2013).

L 229ff: Trends given in this paragraph are not clearly linked to Fig 2a. For example, I do not see that Iceland has a much lower warming than Northern Norway in Fig. 2a. This highlight the need to clarify how trends are calculated.

That is right, Figure 2a shows decadal air temperature trends during the 30-year normal period 1991–2020 based on ERA5 reanalysis data (Hersbach et al., 2020), and thus a product of climate simulations with too coarse resolution to see much differences between Iceland and northern Norway. The weather stations and our data, however, indicate stronger warming in northern Norway.

L263-265: a few explanations about how the vegetation cools the ground would be interesting.

We wrote " … *which cools the ground surface during summer due to shading, and/or a more persistent snow cover during spring, when SAT becomes positive*". I think these points explain the process in short.

L 284: active layer thickness here is mentioned but not related to a figure or data, which makes it difficult to follow the statement contained in this sentence. Reversing 4.3 and 4.4 is maybe a solution to mention ALT when discussing GT.

And

L 310: ice-richness is mentioned but it is not clear how the ice content is determined. This might be worth adding a few word about that in Tab. 1 and sect. 2.

We see your point, but we want to keep the order, first temperatures, then active layer thickness (which of course is also related to temperatures, but also other factors. Instead, we removed the reference to ALT in l. 284. The ice content was not quantified

but inferred during the drilling and related to the amount of sediments covering bedrock. We have rephrased and explained the observations of temperatures and trends in more detail, and the paragraph now reads:

*"In general, ground temperatures (GT) at 10 m depth increased during the measurement period (Figure 3, Table 1), although three cold years in 2010-12 led to a temporary cooling of ground temperatures in southern Norway (Figure 3b). Since then, GT increased in an accelerated pace, and the GT trend at c. 10 m depth varied between 0 and +0.5°C dec$^{-1}$ (Table 1). In northern Norway, a warming trend prevailed during the entire measurement period, with values between +0.4 and +0.5°C dec$^{-1}$ at 10 m depth for all sites. In Iceland, GT trends were also mainly positive, but below +0.3°C dec$^{-1}$ (Table 1). In general, the warmest years has been recorded since 2018 at all sites with the exception of 2021 and 2022 (Figures 3 and 4). The fastest increase of GT after the cool period in 2010-2012 was observed in Tronfjell, southern Norway, possibly because of loss of ground ice, facilitating rapid warming of the ground. Also, the Jetta BH1 site show a somewhat steeper temperature increase. This site is drilled in pure bedrock and has therefor little ice content."*

L323: this is not the talik that thawed, but the permafrost.

Right, we added the word "*permafrost*".

L 356: "Geophysical changes" here means geoelectrical as the seismic data are not presented. Include the seismic data or rephrase throughout the MS to make it clear that this is only geoelectrical data.

We changed to "*Apparent resistivity changes*"

L 409: the statement is interesting, but this needs to be a little bit more detailed. Could you better explain or illustrate how the snow controls GT evolution?

We added following text:

*"Temporal variability in snow cover is an additional driver of changes in ground surface and permafrost temperatures owing to its insulating effect, which restricts winter heat loss from the ground and modulates the influence of air temperature changes on the ground thermal regime (Smith et al. 2022)."*

Fig. 1: report name of the area in 1c. Write boreholes that are not used in the study differently (if I understand well this is for example the case at 1e (Juvflye) as it is stated L83-84 that there are 7 BH but only 5 are used in the study.

Done

Tab. 1: What blue and red texts mean? Those trends are already some results and it is not explained how they are calculated

As mentioned above, we have added an explanation in the table caption. Blue and red denotes below and above 0°C, respectively.

Fig. 2: graphs c and d are barely readable. Make sure that names reported in these graphs are also reported in Fig. 1.

All site names have been mentioned in Figure 1. To make the figures more readable we have selected from (c) and (d) and increased fonts. We made a selection to avoid making two figures.

Fig. 3b: it is interesting to note that at Gu1, permafrost disappeared in 20113-2014 but formed again in 2017 and after. Does this deserve more discussion?

These are 10 m depths measurement, and the BH at Guolasjavri is drilled in pure bedrock (very low water/ice content, very rapid response to forcing) the graph virtually shows that since 2013 the BH is at or very close to 0°C, but in greater depths there is probably still permafrost.

Fig. 4: add a grid in the background of the graphs, especially a line at 0°C.

There was a grid in the background, probably not visible in the draft copy. We add the 0° line in all graphs

Fig. 5: it is interesting to note that the talik at BH2 is reversible.

Yes, as mentioned in the text and Table 1, and in the future Table 2 (model parameters), the bedrock is quartzite which has a very high thermal conductivity. This means some cool winters can penetrate deep in terms of freezing if little snow.

Fig. 7: what are the black lines at the bottom of the graphs? What is the R² between modeled and observed values?

The black bars are daily SWE (numbers at right axis, we change the numbers into black color as the axis title). We added R2-values to the graphs, along with regression and 1:1 lines.